# Brain asymmetries from mid- to late life and hemispheric brain age

Max Korbmacher [1,2,3] ✉, Dennis van der Meer[2,4], Dani Beck [2,5,6],
Ann-Marie G. de Lange[2,7,8], Eli Eikefjord[1,3], Arvid Lundervold[3,9],
Ole A. Andreassen [2,10], Lars T. Westlye [2,6,10] & Ivan I. Maximov [1,2] ✉

The human brain demonstrates structural and functional asymmetries which have implications for ageing and mental and neurological disease development. We used a set of magnetic resonance imaging (MRI) metrics derived from structural and diffusion MRI data in N=48,040 UK Biobank participants to evaluate age-related differences in brain asymmetry. Most regional grey and white matter metrics presented asymmetry, which were higher later in life. Informed by these results, we conducted *hemispheric brain age* (HBA) predictions from left/right multimodal MRI metrics. HBA was concordant to conventional brain age predictions, using metrics from both hemispheres, but offers a supplemental general marker of brain asymmetry when setting left/right HBA into relationship with each other. In contrast to WM brain asymmetries, left/right discrepancies in HBA are lower at higher ages. Our findings outline various sex-specific differences, particularly important for brain age estimates, and the value of further investigating the role of brain asymmetries in brain ageing and disease development.

There are various structural and functional differences in brain architecture between the left and right hemispheres[1–6]. Microstructural brain characteristics, such as white matter (WM) pathways or intra- and extra-neurite water organisation, might underlie the brain's functional lateralisation[7]. Furthermore, handedness has been repeatedly assessed together with asymmetry in humans and animals, and relates to brain asymmetry[8]. Both structural and functional brain asymmetry exhibit clinical importance as there are differences in brain asymmetry between healthy controls and various disease groups, including neurodegenerative diseases such as Alzheimer's disease[9,10], Parkinson's disease[11], and psychiatric disease such as obsessive-compulsive disorder[12,13] and schizophrenia[14]. In that context and particularly relevant from a lifespan-perspective, cortical thickness asymmetry

decreases throughout ageing, with this alteration being potentially accelerated in the development of neurodegenerative disorders such as Alzheimer's Disease[9]. Similarly, some studies suggest lower WM microstructure asymmetry at higher ages, indicated by intra-axonal water fraction[15], fractional anisotropy, or the apparent diffusion coefficient[16]. Additional investigations into brain asymmetries' age-dependencies can provide a more comprehensive understanding of the influence of asymmetries on ageing and disease development.

Brain age is a developing integrative marker of brain health, particularly sensitive to neurodegenerative diseases[17,18]. Brain age refers to the predicted age in contrast to chronological age and is based on a set of scalar metrics derived from brain scans such as MR. To date, brain age has often been estimated using a global brain parametrisation such

¹Department of Health and Functioning, Western Norway University of Applied Sciences, Bergen, Norway. ²NORMENT Centre for Psychosis Research, Division of Mental Health and Addiction, University of Oslo and Oslo University Hospital, Oslo, Norway. ³Mohn Medical Imaging and Visualization Centre (MMIV), Bergen, Norway. ⁴Faculty of Health, Medicine and Life Sciences, Maastricht University, Maastricht, Netherlands. ⁵Department of Psychiatric Research, Diakonhjemmet Hospital, Oslo, Norway. ⁶Department of Psychology, University of Oslo, Oslo, Norway. ⁷Department of Clinical Neurosciences, Lausanne University Hospital (CHUV) and University of Lausanne, Lausanne, Switzerland. ⁸Department of Psychiatry, University of Oxford, Oxford, UK. ⁹Department of Biomedicine, University of Bergen, Bergen, Norway. ¹⁰KG Jebsen Centre for Neurodevelopmental Disorders, University of Oslo, Oslo, Norway. ✉e-mail: max.korbmacher@hvl.no; ivan.maximov@hvl.no

as the averaged scalar measures over particular anatomical regions or the whole brain[17–21]. Hence, we refer to these whole-brain age predictions as global brain age (GBA). However, while brain age has been calculated for different brain regions[18,22–24], the use of hemisphere-specific data is usually not being considered as a potential source of additional information. Yet, one study presents hemisphere-specific and region-specific brain ages containing useful clinical information about post-stroke cognitive improvement[22].

Previous results show that brain age prediction depends on the specific features used[25–27], rendering for example modality as important. Yet, the influence of hemispheric differences or brain asymmetry on the age predictions remains unclear. However, previously outlined brain asymmetries[1–6] might be informative for age predictions. One way of leveraging brain asymmetries into simple metrics is to estimate separate brain ages for each hemisphere (HBA) and to then compare the estimates. It remains unclear whether predictions from a single hemisphere lead to less accurate predictions due to the inclusion of less data and a potential attenuation of noise. At the same time, in the case of diffusion MRI (dMRI), different model-based diffusion features yield highly concordant brain age predictions, also when varying the number of included features[21]. Finally, although the evidence is mixed on the influence of handedness on brain asymmetry[28–31], differences in handedness are potentially reflected in brain structure, which would in turn influence age predictions differently when obtained from the left or right hemisphere only. Hence, handedness requires further examination as potential confounding effect when assessing asymmetry.

HBA, a new brain age measure, may propose more sensitive brain health markers than GBA, as age predictions can be compared between hemispheres to infer the integrity of each hemisphere and give a general estimate of brain asymmetry. Brain asymmetries are commonly observed using the Laterality Index (LI)[32]. However, different ways of estimating asymmetry can introduce variability in its dependency with age[33], and covariates of brain age require further investigation[34,35]. To extend the existing brain age conceptualisation of using features across the whole brain and to maximise interpretability, we restrict brain age predictions to region-averaged and global features and not asymmetries of these features. Additionally, differences in the models' abilities to predict age from WM microstructure features derived from dMRI compared to $T_1$-weighted features (volume, surface area, thickness) need to be ruled out in order to validate both GBA and HBA.

Hence, in the present work, we tested first the preregistered hypotheses (written study and analysis plan prior data inspection and analyses[36,37]) that the GBA and HBA depend on the used MRI modality (Hypothesis 1), disentangling whether the different grey matter (GM) and WM metrics and the degree of their asymmetry influences brain age predictions. We furthermore tested whether there was an effect of hemisphere (Hypothesis 2) and handedness (Hypothesis 3) on brain age predictions. Exploratory analyses included (a) revealing hemispheric differences between GM and WM features, (b) examining LI associations with age, including the LI of the brain features as well as left and right brain ages, and (c) testing the consistency of brain age-covariate associations (specifically, health-and-lifestyle factors, as these were previously associated with brain age[20,26,38–41]).

## Results
### Hemispheric differences and age sensitivity for GM and WM features
Two-tailed paired samples *t*-tests showed that a significant proportion of the GM and WM features differed between hemispheres with medium effect sizes. Among the significant 793 of 840 dMRI feature asymmetries (94.4%, $p_{adj}$< 0.05, with Cohen's $|\bar{d}_{dMRI}|$ = 0.57 ± 0.44). The largest differences were found for DTI FA in the inferior longitudinal fasciculus ($d$ = 3.64), and cingulum ($d$ = 1.95), and for AD in superior longitudinal fasciculus.

Effects sizes of the significant hemispheric differences of the 115 of 117 $T_1$-weighted features (98.3%), were similar: mean $|\bar{d}_{T_1}|$ = 0.53 ± 0.41, and the largest asymmetries were found for the surface area of the transverse-temporal region ($d$ = 1.81), frontal pole ($d$ = 1.76), and pars orbitalis ($d$ = 1.74; see Supplementary Table 10 for $T_1$-weighted and dMRI features with strongest hemispheric differences).

Likelihood Ratio Tests (LRTs) comparing a baseline model predicting age from sex and scanner site compared to a model where the respective smooth of the metric was added (Eq. (3) and (4)) indicated most features as age-sensitive (231 of the 234 (98.72%) of the $T_1$-weighted features; 1601 of the 1680 (95.53%) dMRI features). Age-sensitivity was strongly expressed in both significant $T_1$-weighted features ($\bar{F}_{T_1}$ = 1168.90 ± 993.59), as well as significant dMRI metrics ($\bar{F}_{dMRI}$ = 1208.97 ± 943.52) with strongest age-sensitivity observed for left superior temporal thickness, left/right overall thickness, left/right hippocampus volume, and right inferior parietal thickness and multiple WMM metrics in the right anterior limb of the internal capsule, the left/right fornix-striaterminalis pathway, left/right anterior corona radiata and inferior fronto-occipital fasciculus ($F$ > 3000; for top features see Supplementary Table 2). Results were similar when comparing linear models to the baseline model (Eq. (2) and (4)): 1448 of the 1680 (86.19%) dMRI metrics, and 228 of the 234 (97.44%) of the $T_1$-weighted features were age-sensitive ($\bar{F}_{T_1}$ = 3426.89 ± 2947.11, $\bar{F}_{dMRI}$ = 2378.46 ± 2357.80), with the features with the strongest age-sensitivity resembling LRT results of non-linear models (for top features see Supplementary Table 3).

Considering only left/right averages identified only DTI-AD, and WMTI axial and radial extra-axonal diffusivity to not differ between hemispheres ($p_{adj}$ >0.05). Furthermore, all features were age-sensitive when GAMs ($p_{adj}$ < 3.4 × 10$^{-64}$; yet for linear models, BRIA-vCSF and WMTI-axEAD, as well as right DTI-AD and left WMTI-radEAD were not age sensitive (Supplementary Tables 4, 5). Furthermore, the age-relationships for most of the left/right averages were similar across hemispheres (Fig. 1, both for crude and adjusted values: Supplementary Fig. 1, and for linear and non-linear models: Supplementary Fig. 4). However, differences in dMRI metrics were observed for the ends of the distribution including individuals aged younger than 55 ($N$ = 5307) and older than 75 ($N$ = 3480).

### GM and WM feature asymmetry
Using LRTs comparing GAMs to a baseline model 53 (45.30%) of the 117 $T_1$-weighted and 733 of the 840 (87.26%) dMRI $|LI|$ features as age sensitive ($p_{adj}$< 0.05). Using LRTs on linear effects identified 53 (45.30%) of the 117 $T_1$-weighted and 678 of the 840 (80.71%) dMRI $|LI|$ features as age sensitive ($p_{adj}$< 0.05).

In the following we constrain analyses to linear models and present partial derivatives/slopes as a measure of effect size, allowing for simple comparisons across age-relationships as model fit indices AIC and BIC of linear models and GAMs suggested on average no differences across both $T_1$-weighted ($p_{adj\ AIC}$ = 0.759; $p_{adj\ BIC}$ = 1) and diffusion-weighted features ($d_{AIC}$ = 0.510, $p_{adj\ AIC}$ = 0.020; $p_{adj\ BIC}$ = 0.126).

The absolute feature asymmetries were higher later in life ($\bar{\beta}_{dMRI}$ = 0.05 ± 0.07; $\bar{\beta}_{T_1}$ = 0.03 ± 0.06, $|\bar{\beta}_{multimodal}|$ = 0.05 ± 0.07, here only $p_{adj}$ < 0.05 selected; Supplementary Figs. 2, 3).

The strongest adjusted relationships between the respective features' asymmetries and age were found for dMRI metrics ($|\bar{\beta}_{dMRI}|$ = 0.06 ± 0.04, $|\bar{\beta}_{T_1}|$ = 0.05 ± 0.03; Fig. 2), particularly outlining asymmetry increases in the tapetum ($\beta_{SMTmc-intra}$ = 0.24, $\beta_{BRIA-Vintra}$ = 0.24, $\beta_{SMT-FA}$ = 0.23) and fornix-stria terminalis ($\beta_{DTI-MD}$ = 0.22, $\beta_{BRIA-Vcsf}$ = 0.21), and decrease in the tapetum ($\beta_{SMT-long}$ = −0.23), cerebral peduncle ($\beta_{SMTmc-extratrans}$ = −0.20, $\beta_{SMT-trans}$ = −0.19, $\beta_{BRIA-Vextra}$ = −0.16) and superior longitudinal temporal fasciculus ($\beta_{SMT-long}$ = −0.14).

For $T_1$-weighted metrics, larger, and central structures' $|LI|$ were most sensitive to age, with the strongest negative associations

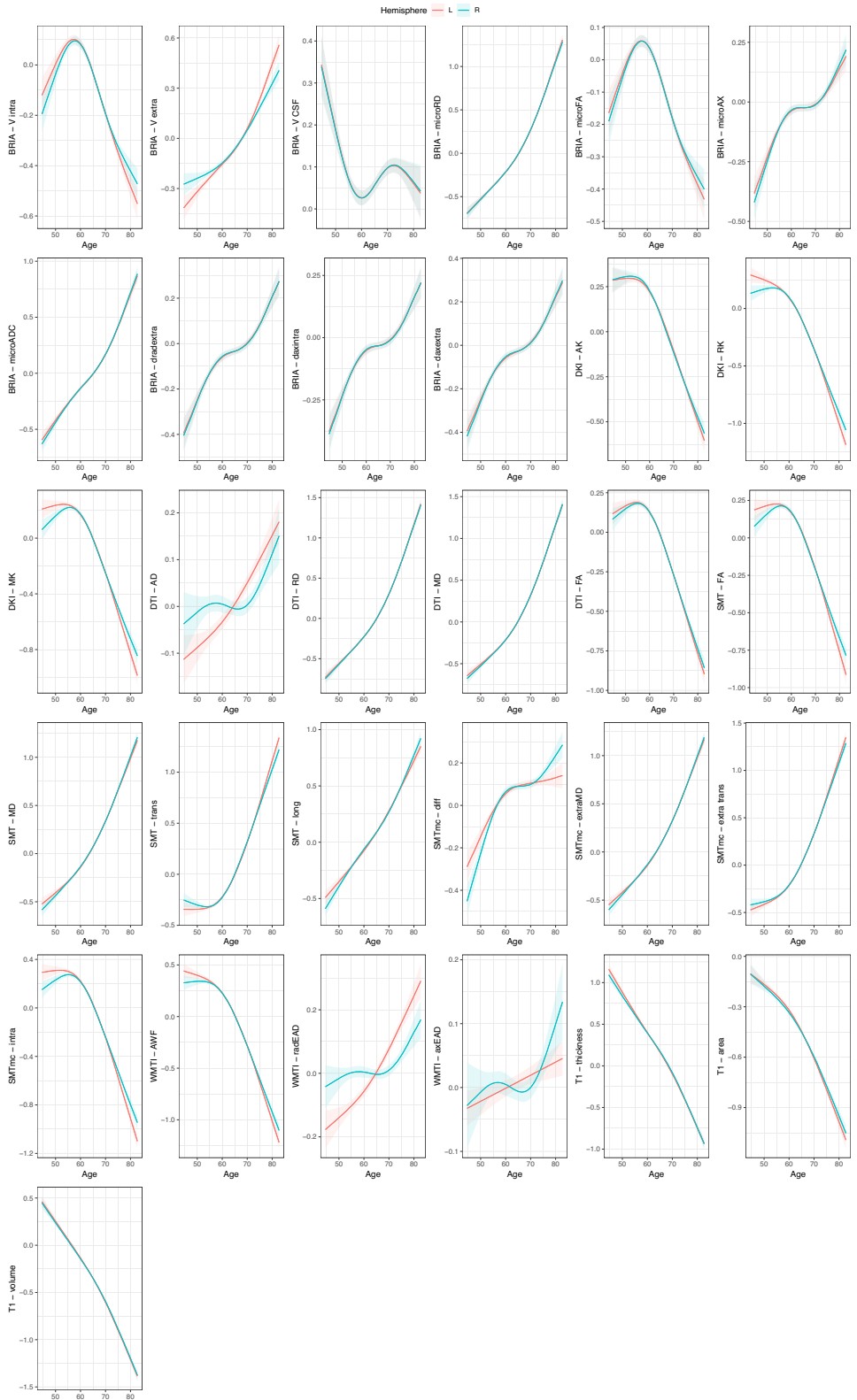

**Fig. 1 | Age curves of standardized and zero-centered mean values of GM and WM features per hemisphere.** A cubic smooth function (s) with $k = 4$ knots was applied to plot the relationship between age and brain features correcting for sex and scanner site (F): $\hat{age} = s(F) + sex + site$ using restricted maximum likelihood (REML). The grey shaded area indicates the 95% CI. All age-relationships were significant ($p_{adj} < 3.4 \times 10^{-64}$). Sample sizes for diffusion metrics: $N_{dMRI} = 39,637$, sample size for $T_1$ metrics: $N_{T_1} = 48,040$. Source data are provided in Source Data file 1.

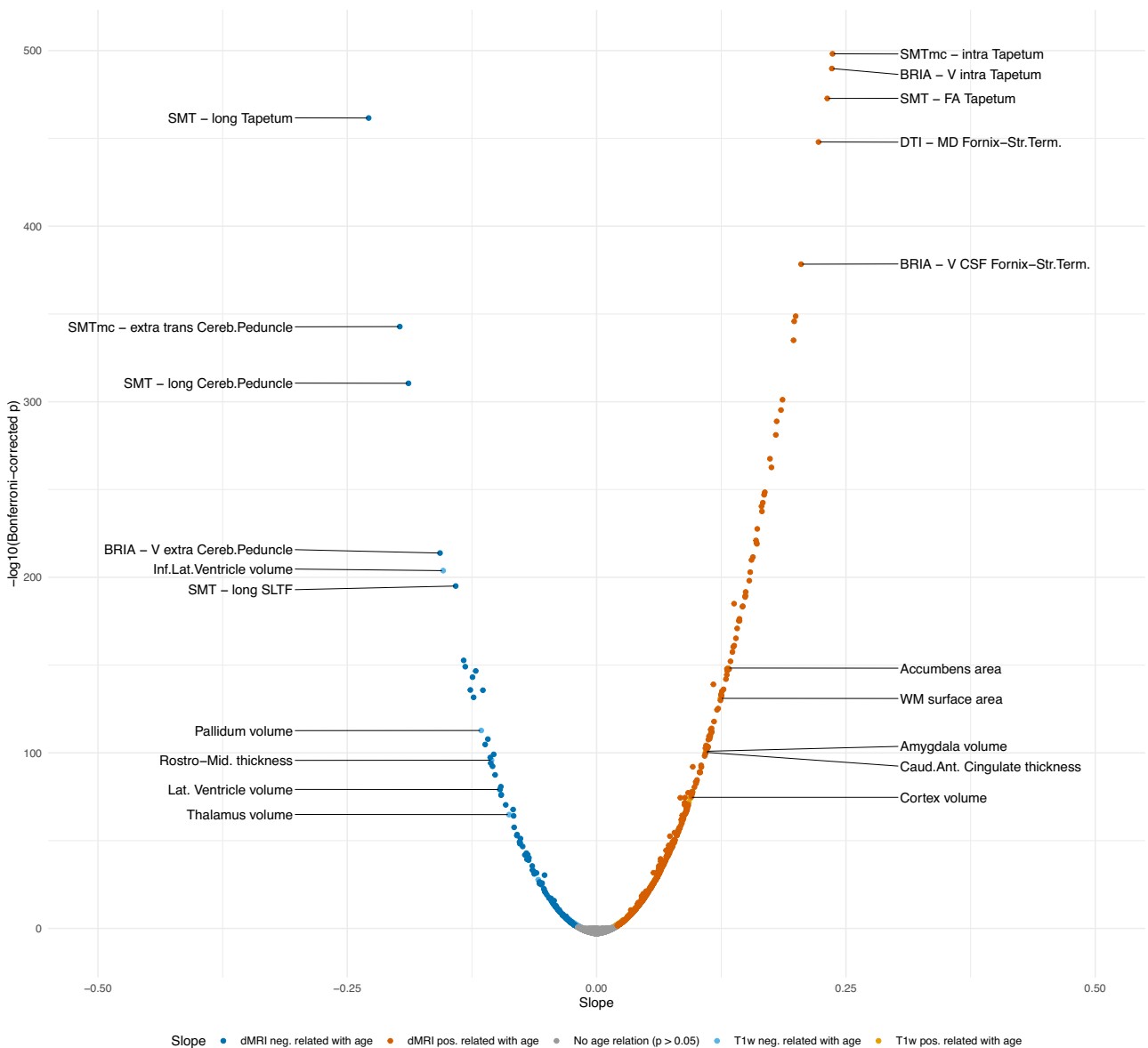

**Fig. 2 | $T_1$-weighted and dMRI features linear asymmetry-age-associations.** The plot presents the standardized (sex- and site-corrected) regression slopes versus Bonferroni-adjusted -log10 $p$-values. Modelling was done using Eq. (2): $a\hat{g}e = \beta_0 + \beta_1 \times F + \beta_2 \times Sex + \beta_3 \times Site$, where $F$ is the respective brain feature. Labelling was done separately for $T_1$-weighted and dMRI indicating the 10 most significantly associated features (five for $\beta > 0$ and five for $\beta < 0$). ILF inferior longitudinal fasciculus, Cereb.Peduncle Cerebral peduncle, Rostro-mid. thickness Rostro-middle thickness, SLFT Superior longitudinal fasciculus (temporal part), Fornix-Str.Term. Fornix-stria terminalis tract, Caud. ant. cingulate Caudal anterior cingulate. Sample sizes for diffusion metrics: $N_{dMRI}$ = 39,637, sample size for $T_1$ metrics: $N_{T_1}$ = 48,040. Source data are provided in Source Data file 2.

including the inferior lateral ($\beta = -0.15$) and lateral ventricles ($\beta = -0.10$), pallidum ($\beta = -0.12$) volumes, rostro-middle thickness ($\beta = -0.11$), thalamus volume ($\beta = -0.09$) and enthorinal area ($\beta = -0.06$). Largest positive age-associations were were shown for accumbens area ($\beta = 0.13$), WM surface area ($\beta = 0.12$), amygdala volume ($\beta = 0.11$), caudal anterior cingulate thicknes ($\beta = 0.11$), cortical ($\beta = 0.09$) and white matter volume ($\beta = 0.09$), as well as caudate volume ($\beta = 0.09$), in addition to several temporal and limbic areas (Fig. 2).

### Sex-specific differences in the influence of hemisphere, modality, and handedness on brain age estimates

Model performance metrics indicated that most accurately age predictions were accomplished using multimodal MRI data based on left, right, and both hemispheres (Table 1), with obtained HBA and GBA being strongly correlated with each other for similar models (Fig. 3).

Additional sex-stratified models produced similar results in terms of model performance (Supplementary Table 14), associations across brain ages and age (Supplementary Fig. 10, and feature importance rankings (compare Supplementary Tables 11–13).

LMERs did not indicate a difference between modalities (Hypothesis 1) when comparing brain ages estimated from both sexes from dMRI to multimodal MRI ($p = 0.623$), and dMRI to $T_1$-weighted MRI ($p = 0.452$). There were also no differences in brain age estimates between hemispheres ($p = 0.413$, Hypothesis 2). Moreover, LRTs indicated no significant difference between models when adding handedness ($\chi^2 = 4.19$, $p = 0.123$, $df = 2$) or handedness-hemisphere interaction and handedness ($\chi^2 = 7.32$, $p = 0.120$, $df = 4$; see Eqs. (5)–(6)).

To additionally consider sex differences, we estimated additional sex-specific brain ages and control for the modelling choice (as extension to Eq. (6)). We find that females' brain ages do not differ when estimated from females' data only compared to predictions from

**Table 1 | Hemispheric brain age prediction outcomes**

| Model | Features | $R^2$ | MAE | RMSE | Correlation* |
|---|---|---|---|---|---|
| Left T$_1$w | 117 | 0.504 (0.010) | 4.389 (0.054) | 5.472 (0.061) | 0.708 [0.703, 0.712] |
| Right T$_1$w | 117 | 0.492 (0.008) | 4.439 (0.049) | 5.529 (0.051) | 0.705 [0.700, 0.709] |
| T$_1$w | 234 | 0.526 (0.011) | 4.294 (0.050) | 5.356 (0.062) | 0.725 [0.721, 0.730] |
| Left dMRI | 840 | 0.568 (0.014) | 4.000 (0.047) | 4.990 (0.067) | 0.757 [0.753, 0.762] |
| Right dMRI | 840 | 0.582 (0.013) | 3.960 (0.052) | 4.967 (0.079) | 0.766 [0.762, 0.771] |
| dMRI | 1680 | 0.605 (0.010) | 3.867 (0.059) | 4.821 (0.094) | 0.781 [0.777, 0.785] |
| Left multimodal | 957 | 0.630 (0.009) | 3.757 (0.046) | 4.673 (0.047) | 0.794 [0.790, 0.797] |
| Right multimodal | 957 | 0.634 (0.014) | 3.723 (0.073) | 4.673 (0.092) | 0.794 [0.791, 0.798] |
| Multimodal | 1914 | 0.628 (0.017) | 3.663 (0.055) | 4.563 (0.077) | 0.793 [0.789, 0.797] |

$R^2$ Variance explained, *MAE* Mean Absolute Error, *RMSE* Root Mean Squared Error, *Corr.* Correlation, Values in round parentheses () refer to standard deviations and square brackets [] to 95% confidence interval around correlations (Pearson's r) of uncorrected brain age estimates and chronological age. *N* = 35,665. Source data are provided in Source Data file 5.
*The correlation between raw brain age and chronological age. All $p_{adj}$ < 0.001.

Correlations between predicted and chronological age

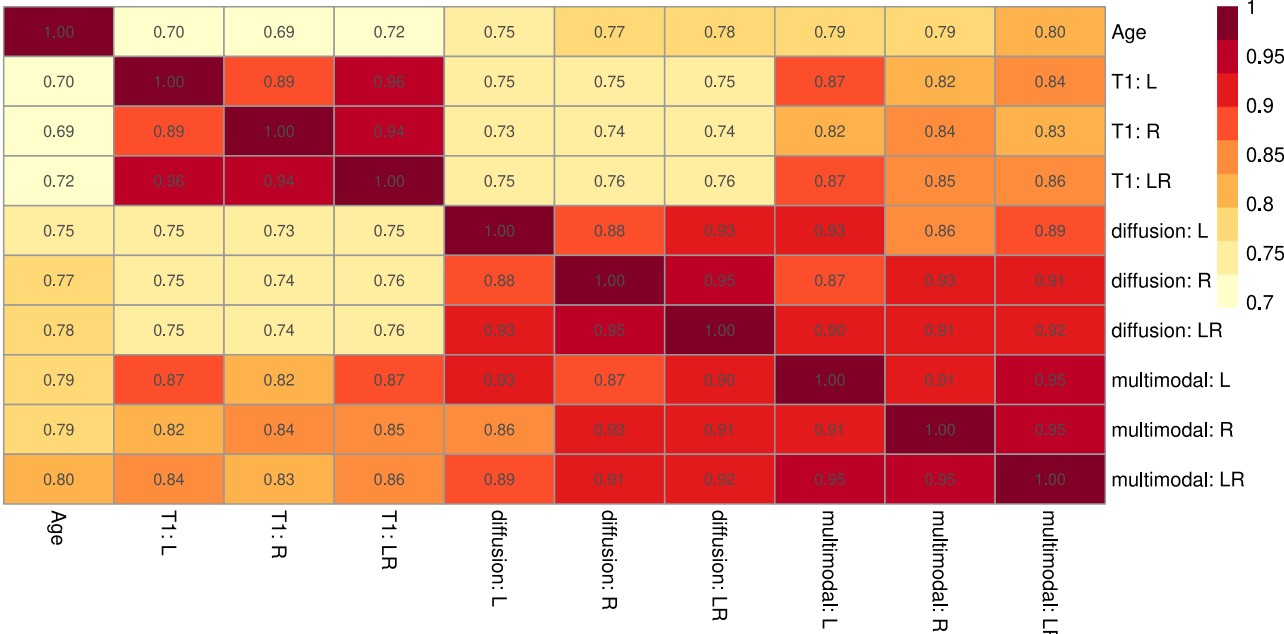

**Fig. 3 | Pearson correlation coefficients between chronological and predicted ages for T$_1$-weighted, diffusion, and multimodal MRI for left, right and both hemispheres, *N* = 35,665.** All Bonferroni-corrected *p* < 0.001. L Left hemisphere, R Right hemisphere, LR Both hemispheres. Source data are provided in Source Data file 3.

both males' and females' data ($\beta = -0.0073$ years, $p = 0.420$). The same holds true for male brain ages estimated from males' data only compared to data from both sexes ($\beta = -0.0002$ years, $p = 0.984$). Furthermore, with these additional modelling choices, we identified a significant marginal effect of sex (indicating an older brain age for males: $\beta = 0.58$ years, $p < 0.001$), and hemisphere for T$_1$-weighted ($\beta = 0.03$ years, $p = 0.022$), but not dMRI ($\beta = 0.02$ years, $p = 0.099$), or multimodal MRI ($\beta = 0.02$ years, $p = 0.110$). Moreover, ambidextrous brain age was higher than for left-handed ($\beta = 1$ year, $p < 0.001$) and right handed participants ($\beta = 0.7$ years, $p < 0.001$), as well as higher for right-handed compared to left-handed participants ($\beta = 0.2$ years, $p < 0.001$).

Further investigating the identified sex-effect, we found higher brain ages for males across modalities with larger differences identified for dMRI ($\beta_{left} = 0.768$ years, $p < 0.001$, $\beta_{right} = 0.870$ years, $p < 0.001$), followed by T$_1$-weighted ($\beta_{left} = 0.308$ years, $p < 0.001$, $\beta_{right} = 0.438$ years, $p < 0.001$) and multimodal MRI ($\beta_{left} = 0.503$ years, $p < 0.001$, $\beta_{right} = 0.570$ years, $p < 0.001$). Notably,

females' right brain age was lower than the left brain age ($\beta_{T_1} = -0.035$ years, $p = .027$, $\beta_{dMRI} = -0.029$ years, $p = 0.066$, $\beta_{multimodal} = -0.013$ years, $p = 0.403$), which was the opposite for males showing lower left brain age ($\beta_{T_1} = 0.095$ years, $p < .001$, $\beta_{dMRI} = 0.073$ years, $p < 0.001$, $\beta_{multimodal} = -0.054$ years, $p = 0.001$). In contrast to the analyses across sexes, these additional analyses provide support for Hypotheses 1-3 when sex-stratifying.

**Lower brain age asymmetry at higher ages**

To test whether asymmetries between hemisphere-specific brain age predictions are lower at higher age, $|LI_{HBA}|$, was associated with age (Eq. (7)–(8)). $|LI_{HBA}|$ showed negative unadjusted associations with age for T$_1$-weighted ($r = -0.069$, $p < 0.001$), dMRI ($r = -0.121$, $p < 0.001$), and multimodal models ($r = -0.121$, $p < 0.001$). The associations were similar when using LMEs adjusting for sex and the random intercept site (T$_1$-weighted: $\beta = -0.069$, $p < 0.001$, dMRI: $\beta = -0.115$, $p < 0.001$, multimodal: $\beta = -0.117$, $p < 0.001$). LRTs indicate the age-sensitivity of $LI_{HBA}$ (T$_1$-weighted:

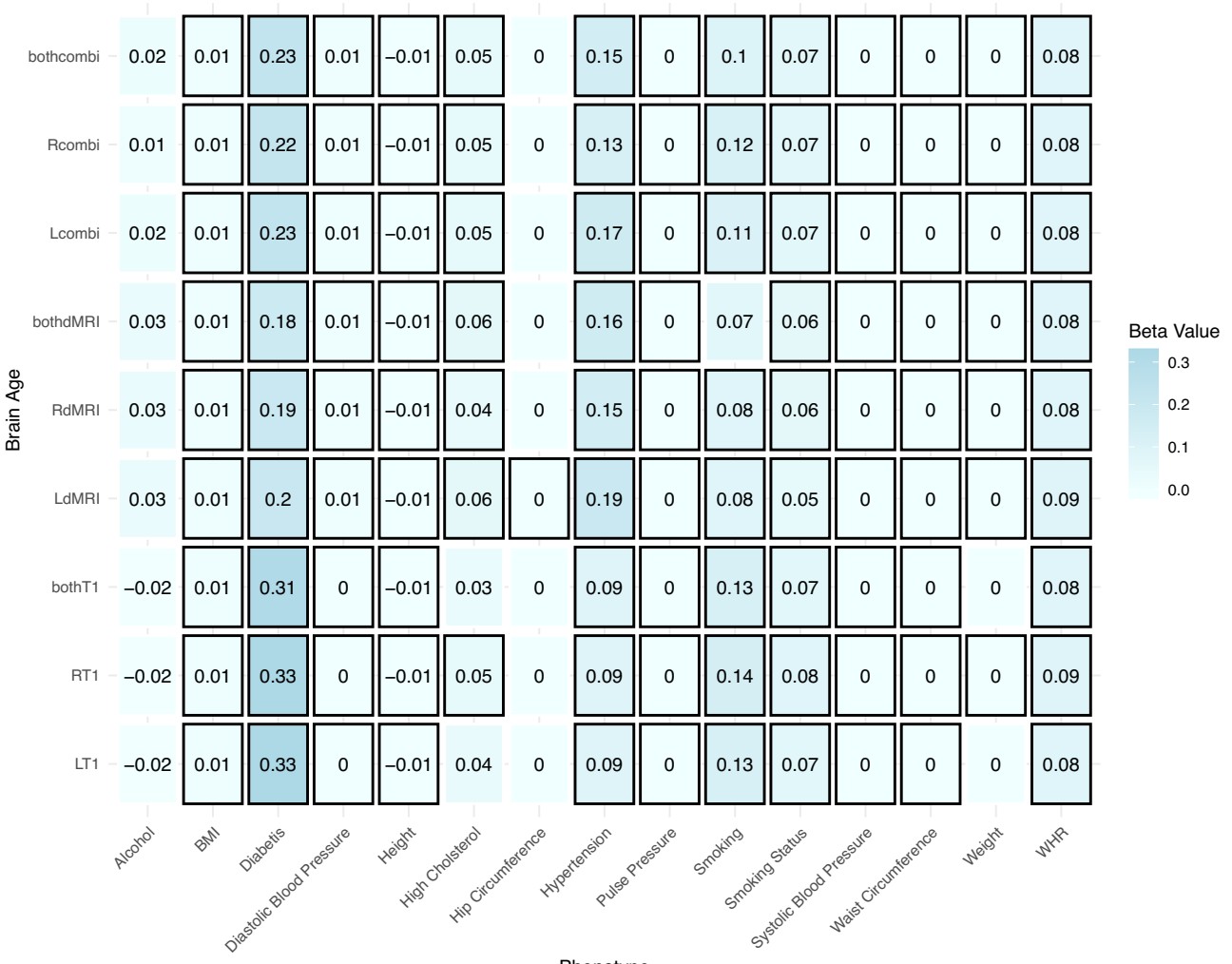

**Fig. 4 | Linear association between general health-and-lifestyle phenotypes and brain age estimated from different modalities, left, right and both hemispheres.** Eq. (9) was used and standardized slopes are presented. For simplicity, standardized slopes with $|\beta| < 0.005$ were rounded down to $\beta = 0$. L Left hemisphere, R right hemisphere, LR Both hemispheres, BMI Body mass index, WHR Waist-to-hip ratio. Bonferroni-adjusted $p < 0.05$ is marked by a black frame. Sample size: $N = 35,665$. Missingness in predictors: $N_{Alcohol} = 237$, $N_{BMI} = 1247$, $N_{Diastolic\ Blood\ Pressure} = 4442$, $N_{Height} = 1189$, $N_{Hip\ Circumference} = 1153$, $N_{Pulse\ Pressure} = 4442$, $N_{Smoking} = 339$, $N_{Smoking\ Status} = 224$, $N_{Systolic\ Blood\ Pressure} = 4442$, $N_{Waist\ Circumference} = 1154$, $N_{Weight} = 1212$, $N_{WHR} = 1154$. Source data are provided in Source Data file 4.

$\chi^2 = 173.42$, $p < 0.001$, dMRI: $\chi^2 = 488.74$, $p < 0.001$, multimodal: $\chi^2 = 506.08$, $p < 0.001$).

These results were robust to stratifying by sex, estimates from a brain age model considering both sexes for unadjusted ($r_{dMRI\ males} = -0.134$, $r_{dMR\ lfemales} = -0.104$, $r_{T1\ males} = -0.134$, $r_{T1\ females} = -0.048$, $r_{multimodal\ males} = -0.134$, $r_{multimodal\ females} = -0.111$), and adjusted associations ($\beta_{dMRI\ males} = -0.134$, $\beta_{dMRI\ females} = -0.099$, $\beta_{T1\ males} = -0.134$, $\beta_{T1\ females} = -0.045$, $\beta_{multimodal\ males} = -0.134$, $\beta_{multimodal\ females} = -0.106$), with $\chi^2$ tests suggesting age sensitivity (all $p < 0.001$).

Using brain age predictions from models which were independently estimated for males and females showed similar results for unadjusted ($r_{dMRI\ males} = -0.141$, $r_{dMRI\ females} = -0.094$, $r_{T1\ males} = -0.120$, $r_{T1\ females} = -0.031$, $r_{multimodal\ males} = -0.165$, $r_{multimodal\ females} = -0.089$), and adjusted associations ($\beta_{dMRI\ males} = -0.137$, $\beta_{dMRI\ females} = -0.088$, $\beta_{T1\ males} = -0.117$, $\beta_{T1\ females} = -0.029$, $\beta_{multimodal\ males} = -0.162$, $\beta_{multimodal\ females} = -0.084$), with $\chi^2$ tests suggesting age sensitivity (all $p < 0.001$).

Finally, also when analysing brain ages for males and females from sex-specific models together shows similar trends for uncorrected $|LI_{HBA}|$-age associations ($r_{multimodal} = -0.123$, $p < 0.001$; $r_{T1} = -0.074$, $p < 0.001$, $r_{dMRI} = -0.114$, $p < 0.001$), as well as corrected association

($\beta_{multimodal} = -0.125$, $p < 0.001$; $\beta_{T1} = -0.071$, $p < 0.001$, $\beta_{dMRI} = -0.113$, $p < 0.001$; Eq. (7)–(8)).

### HBA and GBA and health-and-lifestyle factors

We further investigated the pattern of relationships with general health-and-lifestyle phenotypes across HBAs (Fig. 4). Relationships between brain ages from single and both hemispheres were similar within modalities, but varied slightly between modalities (Fig. 4). These results were robust to sex stratifications. Yet, while males' brain age was sensitive to high cholesterol, hip circumference, smoking and weight, this was not the case for females' brain age when using brain age predictions from data of both sexes (Supplementary Figure 11–12).

### Sex stratified hemispheric differences and age sensitivity for GM and WM features

For further insights into sex differences, we repeated the presented analyses on hemispheric differences and features' age-sensitivity stratifying by sex. Two-tailed paired samples $t$-tests assessing regional differences between hemispheres showed similar results between sexes, which are also comparable to cross-sex results. Most features differed between hemispheres for both males and females ($T_1$-

weighted: 98.3% for both sexes, dMRI$_{males}$: 96%, dMRI$_{females}$: 95%), and effect sizes were similar ($|\bar{d}_{T_1males}|$ = 0.54 ± 0.42, $|\bar{d}_{T_1females}|$ = 0.53 ± 0.42, $|\bar{d}_{dMRImales}|$ = 0.57 ± 0.41, $|\bar{d}_{dMRIfemales}|$ = 0.60 ± 0.47).

Also the strongest effects were similar across sexes: strongest differences in T$_1$-weighted features in males were observed for frontal pole ($d_{T_1males}$ = 1.82) and pars orbitalis ($d_{T_1males}$ = 1.78) surface area, and for females in the area of the transverse temporal area ($d_{T_1females}$ = 1.89) and the frontal pole ($d_{T_1females}$ = 1.73). Strongest WM differences were observed for both sexes in inferior longitudinal fasciculus ($d_{dMRI\ males}$ = 3.44, $d_{dMRI\ females}$ = 3.91), and superior longitudinal temporal fasciculus ($d_{dMRI\ males}$ = 2.09, $d_{dMRI\ females}$ = 2.40; Supplementary Table 6).

LRTs comparing a baseline model predicting age from sex and scanner site compared to a model where the respective smooth of the metric was added (Eq. (3) and (4)) indicated most features as age-sensitive (230 of the 234 (98.29%) of the T$_1$-weighted features (both sexes); 1557 and 1564 of the 1680 (92.68% and 93.10%) dMRI features for males and females, respective). Age-sensitivity was strongly expressed in both significant T$_1$-weighted features ($\bar{F}_{T_1males}$ = 640.80 ± 521.33; $\bar{F}_{T_1females}$ = 578.61 ± 500.79), as well as significant dMRI metrics ($\bar{F}_{dMRImales}$ = 586.38 ± 450.68, $\bar{F}_{dMRIfemales}$ = 674.61 ± 499.58).

Similar to the results including both sexes, the strongest T$_1$-weighted feature age-sensitivity was observed for left superior temporal thickness, left/right hippocampus volume for both sexes, and right inferior parietal thickness only for females. Concerning dMRI features, sex stratification reflects the findings accounting for sex, outlining the fornix-striaterminalis pathway, anterior corona radiata and inferior fronto-occipital fasciculus, yet adding the anterior limb of the internal capsule and the anterior thalamic radiation. Unique to non-linear models, also the lateral ventricle volume was lined out as highly age sensitive (all $F > 1666$; for top features see Supplementary Table 7).

Results were similar when comparing linear models to the baseline model (Eq. (2) and (4)): 1557 and 1564 of the 1680 (92.68%, 93.01%) dMRI metrics, and 226 and 224 of the 234 (96.58%, 95.73%) of the T$_1$-weighted features were age-sensitive for males and females, respectively ($\bar{F}_{T_1males}$ = 1767.60 ± 1474.69; $\bar{F}_{T_1females}$ = 1712.73 ± 1488.97; $\bar{F}_{dMRImales}$ = 1198.85 ± 1135.84, $\bar{F}_{dMRIfemales}$ = 1297.51 ± 1257.02), with the features with the strongest age-sensitivity resembling LRT results of non-linear models (for top features see Supplementary Table 8).

Considering only left and right hemispheric averages, $t$-tests indicated that all features differed between hemispheres for males ($p_{adj} < 3.1 \times 10^{-9}$). In females, WMTI radEAD and axEAD as well as DTI AD did not differ between hemispheres ($p_{adj} > 0.05$), but all other metrics differing between hemispheres ($p_{adj} < 1.5 \times 10^{-36}$).

Considering all regional features, LRTs on GAMs (Eq. (4), (3)) indicated that all features were age-sensitive ($p_{adj} < 5.1 \times 10^{-71}$). LRTs on linear models (Eq. (2), (4)) indicated that right hemisphere BRIA-vCSF and left microRD were not age sensitive ($p_{adj} > 0.05$) in males. In females, additionally, left DTI-RD and GM thickness as well as left and right WMTI-axEAD were not age-sensitive. All other metrics were age sensitive ($p_{adj} < 2.7 \times 10^{-11}$). Hemispheric features' age-relationships showed similar intercepts and slopes across sexes, except DTI-AD, WMTI-radEAD and WMTI-axEAD (Supplementary Figs. 5, 6).

### Sex differences in GM and WM feature asymmetry

Sex-stratified analyses indicate most dMRI $|LI|$ features to be age sensitive (dMRI$_{males}$ = 64.29%, dMRI$_{females}$ = 69.52%), but less T$_1$-weighted features (T$_{1\ males}$ = 47.86%, T$_{1\ females}$ = 38.46%) when using non-linear models. Linear models showed similar results (dMRI$_{males}$ = 60.95%, dMRI$_{females}$ = 64.05%; T$_{1\ males}$ = 44.44%, T$_{1\ females}$ = 37.61%). Comparing linear to non-linear models using two-sided paired samples $t$-tests suggests no differences model fit indicated in AIC or BIC scores for both males and females in T$_1$-weighted and diffusion features' asymmetry ($p_{adj} > 0.05$). Hence, linear model outcomes are presented below. Similar to models including both sexes, when stratifying for sex, $|LI|$ for diffusion and T$_1$-weighted feature were positively associated with age ($\bar{\beta}_{dMRImale}$ = 0.05 ± 0.08, $\bar{\beta}_{dMRIfemale}$ = 0.05 ± 0.08, $\bar{\beta}_{T1male}$ = 0.03 ± 0.06, $\bar{\beta}_{T1female}$ = 0.03 ± 0.06).

The strongest adjusted relationships for diffusion features were found in the cingulate gyrus tract ($\beta_{males\ BRIA-microRD}$ = 0.25, $\beta_{males\ BRIA-microFA}$ = 0.22, $\beta_{females\ BRIA-microRD}$ = 0.25, $\beta_{females\ BRIA-microFA}$ = 0.21) and in the cerebral peduncle ($\beta_{males\ SMTmc-extratrans}$ = −0.19, $\beta_{males\ SMT-trans}$ = −0.18, $\beta_{females\ SMTmc-extratrans}$ = −0.21, $\beta_{females\ SMT-trans}$ = −0.20, $\beta_{females\ BRIA-Vextra}$ = −0.18; Supplementary Figs. 8, 9). Strongest age associations with T$_1$-weighted asymmetries were found for the area of the accumbens ($\beta_{males}$ = 0.14, $\beta_{females}$ = 0.12) and WM surface ($\beta_{males}$ = 0.13, $\beta_{females}$ = 0.12), with strongest inverse relationships observed for inferior lateral ventricles ($\beta_{males}$ = −0.17, $\beta_{females}$ = −0.14) and pallidum ($\beta_{males}$ = −0.11, $\beta_{females}$ = −0.12).

## Discussion

In the present work we investigated a new way of utilising brain age to differentiate between hemispheres, and performed a detailed assessment of brain asymmetry associations with age. As a baseline, we showed that most grey and white matter features were age-sensitive and differed between hemispheres with relatively large effect sizes. Brain asymmetry was age-sensitive, and overall higher at higher ages. In contrast, asymmetry in hemispheric brain age was lower at higher ages. The strongest relationship of age and absolute brain asymmetry was identified in larger GM and WM regions, as well subcortical and lower structures, including the limbic system, the ventricles, cingulate and cerebral as well as cerebral peduncle WM.

Brain age predictions exhibited concordant accuracy within modalities for left, right, and both hemispheres, and concordant associations with health-and-lifestyle factors also when analysing data for males and females separately, training brain age models on data from each sex separately or both sexes together. The predictions did not differ statistically between hemispheres, modalities, or handedness groups when considering both sexes together. However, sex-stratified analyses, which considered different brain age modelling choices, revealed significant opposing effects between sexes for hemisphere and modality, and outlined marginal differences between handedness groups. There are multiple reasons for the observed higher brain age in females' right hemisphere compared to males' higher brain age of the left hemisphere, in addition to modality-specific differences. First, male and female brain structure differs, resulting in sex-specific regional variations in brain age estimates[42]. Second, body and brain ageing trajectories differ between sexes, for example, outlined by sex-dependent importance of cardiometabolic risk factors[43]. Hence, the tendency of males' predicted brain age being lower using T$_1$-weighted and multimodal in contrast to diffusion-derived brain ages, with these trends reversed in females, might also reflect stronger brain age associations with cardiometabolic risk factors in males (Supplementary Fig. 7), which have been demonstrated earlier for WM features and WM brain age[38,39]. HBA allows to assess the structural integrity of each hemisphere individually, and to set brain ages from the two hemispheres in relationship to each other providing a general marker of asymmetry. Despite brain asymmetries overall increasing (Supplementary Figs. 2, 3), the asymmetries between left/right HBA were smaller at a higher age. At higher ages, both hemispheres might hence become overall more comparable, despite ageing-related changes[44].

We found that the majority of regional and hemisphere-averaged MRI features differed between hemispheres. Both features and asymmetries were age-sensitive indicating that the investigation of asymmetries are useful across ages and MRI modalities.

Interestingly, hemisphere-averaged features' age-associations and HBA of the same modality were similar between hemispheres (Fig. 1), and the hemisphere was not a significant predictor of brain age estimated from a particular hemisphere, when analysing data from both sexes together. However, when sex-stratifying, modality and hemisphere were significant predictors, suggesting that HBA captures both brain asymmetries as well as biological sex-differences which become apparent when using multimodal MRI. These results outline the importance of considering sex-differences in brain age analyses.

Several studies present evidence for asymmetries in WM[6,45–48] and GM[4,9,49–51]. In contrast to these previous studies, for the first time, we examine various metrics supplying information on both WM and GM in a large sample. While we find various differences between hemispheres, age relationships of $T_1$-weighted and dMRI features were similar between hemispheres using hemispheric averages, also when stratifying by sex. Spatially finer-grained examinations revealed more specific patterns of asymmetry in $T_1$-weighted features, such as GM thickness[9], and dMRI features[45]. This is also shown in the present study by stronger age-effects for specific regional asymmetries compared to asymmetries in hemispheric averages. Age-MRI metric relationships depend, however, on the selected metric, the sample, and the sampling (cross-sectional or longitudinal)[52,53]. For example, previous evidence from $T_1$-weighted MRI indicates no differences in GM volume between hemispheres[54], but hemispheric differences of cortical thickness and surface area across ageing[4,9].

The presented age charts of MRI metrics in the current work (Fig. 1, Supplementary Fig. 1) provide similar trends to those reported in previous studies observing global age dependencies[19,21,55–57]. Yet, the stratification between hemispheres when presenting brain features' age dependence is a novel way of presenting brain charts.

We found asymmetries based on GM and WM brain scalar measures. Unimodal studies with smaller, younger samples presented age-dependence of the brain asymmetry during early WM development[48] and adult cortical thickness[9], other $T_1$-derived metrics[33], and functional network development[5], showing lower asymmetry at higher ages. In contrast to HBA asymmetries, brain asymmetries do generally not support the notion of lower but instead of higher brain asymmetry later in life. Different study design choices, such as temporal and spatial levels might provide supplemental information into the age-dependence of brain asymmetries, for example, by further investigating longitudinal and voxel-level asymmetries.

We extended previous findings by providing a comprehensive overview of brain asymmetry associations throughout mid- to late life including both GM and WM. Our findings indicate that when considering various metrics, older brains generally appear less symmetric than younger brains in the current sample mid- to late-life sample, whereas brain age appears more symmetric in older brains.

Notably, we identified strong associations between specific brain regions' asymmetry and age. The strongest age associations of asymmetries were observed for subcortical, ventricle-near structures. The general age-sensitivity of such structures[21,58,59] might be a reason for the observed age associations in asymmetries, and hence pointing towards one hemisphere being stronger affected by degradation effects, or even the involvement of such regions in psychiatric and neurodegenerative disorders[40,55,58,60–65]. For example, the hippocampus, a prominent limbic structure, presents relatively high levels of adult neurogenesis, which might potentially explain repeated findings of the region's associations with psychiatric disorders and disorder states such as depression, anxiety, schizophrenia, addiction, and psychosis[66,67], and neurdegenerative disorders, especially Alzheimer's Disease[68], but also ageing in general[69]. Some of the strongest age-relationship for $T_1$-derived asymmetries were observed in the accumbens, ventricles and pallidum. In turn, a series of dMRI approaches was sensitive to asymmetry in the cingulum tract, which is higher in late-life and cerebral peduncle asymmetry which appears lower in late-life. In

particular, radial diffusivity metrics, such SMT-trans, SMTmc-extra-trans, and BRIA-microRd, and fractional anisotropy indicated by BRIA-microFA were sensitive to age-dependencies of these asymmetries. Although speculative, this observation could indicate a relationship between asymmetry and axonal properties during ageing, such as myelination, density, or diameter, in the cingulum, with yet a more general marker (BRIA-microFA) of anisotropy asymmetry increasing at advanced age. However, limitations of the different diffusion metrics, such as the inability to account for axonal swelling, infection, or crossing fibres[70], aggravates the interpretation of such asymmetry changes. Overall, asymmetries' age-dependencies in subcortical, limbic and ventricle-near areas are not surprising, considering that the cingulum and cerebral peduncle WM, and middle temporal GM area also presented some of the strongest asymmetries across the sample (Supplementary Table 10).

Both GM volume, surface, and thickness show asymmetries across studies[1,3,4,9,54]. We identified lower asymmetry linked to higher ages in the ventricular and pallidum volumes, appearing alongside the known effect of larger ventricle volumes at higher ages[55]. The strongest positive age-relationships for $T_1$-weighted features' asymmetry were observed for accumbens and WM surface area, as well as limbic structures such as amygdala, hippocampus, and cingulate. Limbic structures have previously been outlined as highly age-sensitive[21,58,59,69]. Higher asymmetry-levels might speak to asymmetric atrophy in these limbic regions, potentially explaining several ageing-related effects[9]. However, lifespan changes in ventricular volume asymmetry in relation to symptom and disorder expression requires additional investigations.

Cingulum WM microstructure has been reported to differ between hemispheres[71–73]. Abnormalities in cingulum asymmetry have been linked to schizophrenia[74–76] and epilepsy[77,78], and Alzheimer's disease[59]. Additionally, the cingulum tract was associated with the anti-depressant effects of deep brain stimulation in treatment-resistant depression[79]. Recent evidence points out strongest polygenic risk associations for several psychiatric disorders in addition to Alzheimer's Disease with longitudinal WM in the cerebral peduncle[58]. Future research could assess regional asymmetries to evaluate such metrics' value for diagnostics and treatment in a range of brain disorders.

Overall, most absolute MRI feature asymmetries were positively related to age, with brain age asymmetries showing inverse age-relationships. However, for both WM and GM this process was observed to be spatially distributed. Metric-specific changes might indicate accelerated and pathological ageing[9], which urges to examine different WM and GM metrics across temporal and spatial resolutions and in clinical samples.

Informed by the presented brain asymmetries and their age-dependence, we suggest HBA, indicating the structural integrity of each hemisphere when compared to the chronological age. Moreover, HBA provides a general marker of asymmetry, when setting left/right HBA in relationship to each other. While this added information to conventional GBA is promising, first, the degree to which HBA captures GBA predictions, had to be assessed. This investigation included (1) direct comparisons of HBA and GBA models and their predictions, (2) the influence of covariates of brain age including MRI modality, hemisphere, handedness, and the hemisphere-handedness interaction effect, and (3) a comparison of health-and-lifestyle phenotype-associations with HBA and GBA. Overall, HBA and GBA were highly similar across these dimensions, yet different between hemispheres and modalities within males and females, with these differences contrasting each other. This renders HBA sensitive to potential underlying biological processes which only become apparent when assessing males and females separately. Additionally, different modalities might be sensitive to a range of biological phenomena in terms of brain age, such as dMRI brain age which presents group differences for diabetes only in males. In that sense, a further route of investigation could be

to establish sex-specific uni- and multimodal brain age models (which account for sex differences in brain morphology and its developmental trajectories). The influence of hemisphere and sex on how these models relate to biological phenomena can then be assessed.

Congruently with previous research that combined MRI modalities[27], we found higher prediction accuracy for multimodal compared to unimodal predictions for both HBA and GBA. Our results extend previous findings on conventional brain age by not only estimating brain age from different MRI modalities, but also for each hemisphere and sex separately. HBA could hold potential in clinical samples by informing about the consistency between the two hemispheres' brain age predictions. Particularly diseases or conditions which affect a single hemisphere, such as unilateral stroke or trauma, might then be sensitively detected, and the integrity of the unaffected hemisphere can be assessed by observing the congruence of HBA[22]. Larger discrepancies between HBAs of the same individual might act as a marker of hemisphere-specific brain health imbalance, which may indicate potential pathology.

While this study provides initial explorations of asymmetries and HBA, our findings remain limited to the examined sample (imaging subset of the UKB), and limited by generational effects within the sample. The UKB contains individuals born in different decades, which influences individual predispositions for brain health through various factors such as the living environment[80] or education[81], representing various potential confounding effects. Additional bias might have been introduced by the sample characteristics and sampling procedure. The UKB consists of nearly exclusively white UK citizens, limiting the generalisability beyond white Northern Europeans and US Americans in their midlife to late life. The volunteer-based sampling procedure might additionally have introduced bias, reducing generalisability to the UK population[82], with the imaging sample of the UKB showing an additional positive health bias (better physical and mental health) over the rest of the UKB sample[83], rendering this sub-sample as even less representative of the total UK population. Finally, the selection of the parcellation scheme for both grey and white matter, and the inherent parcellation bias, might have influenced the results. As a consequence, the reported findings might be more reflective of the parcellation bias than the inherent brain organisation.

In conclusion, we identified asymmetries throughout the brain from midlife to late-life. These asymmetries appear higher later in life across GM and WM. Opposing, the difference in left/right hemispheric brain age is smaller at higher ages. We furthermore identify various sex-specific differences in brain age and its correlates, as well as regional asymmetries which do not only show age-dependence but which have also been related to various clinical diagnoses. The identified age-relationships of asymmetries provide future opportunities to better understand ageing and disease development.

## Methods
### Sample characteristics
We obtained UK Biobank (UKB) data[84], including $N = 48,040$ $T_1$-weighted datasets, $N = 39,637$ dMRI datasets, resulting in $N = 39,507$ joined/multimodal datasets after exclusions were applied. Participant data were excluded when consent had been withdrawn, an ICD-10 diagnosis from categories F (presence of mental and behavioural disorder), G (disease of the nervous system), I (disease of the circulatory system), or stroke was present, and when datasets were not meeting quality control standards using the YTTRIUM method[85] for dMRI datasets and Euler numbers were larger than 3 standard deviations below the mean for $T_1$-weighted data[86]. In brief, YTTRIUM[85] converts the dMRI scalar metric into 2D format using a structural similarity[87,88] extension of each scalar map to their mean image in order to create a 2D distribution of image and diffusion parameters. These quality

assessments are based on a 2-step clustering algorithm applied to identify subjects located outside of the main distribution.

Data were collected at four sites, with the $T_1$-weighted data collected in Cheadle (58.41%), Newcastle (25.97%), Reading (15.48%), and Bristol (0.14%). Of these data, 52.00% were females, and the participants age range was from 44.57 to 83.71, mean = 64.86 ± 7.77, median = 65.38 ± 8.79. DMRI data were available from four sites: Cheadle (57.76%), Newcastle (26.12%), Reading (15.98%), and Bristol (0.14), with 52.19% female, and an age range of 44.57–82.75, mean = 64.63 ± 7.70, median = 65.16 ± 8.73. The multimodal sample ($N = 39,507$) was 52.22% female, with an age range of 44.57–82.75, mean = 64.62 ± 7.70, median = 65.15 ± 8.73. Information on sex was acquired from the UK central registry at recruitment, but in some cases updated by the participant. Hence the sex variable may contain a mixture of the sex the UK National Health Service (NHS) had recorded for the participant as well as self-reported sex.

### MRI acquisition and post-processing
UKB MRI data acquisition procedures are described elsewhere[84,89,90] and can be found at https://www.fmrib.ox.ac.uk/ukbiobank/protocol/. The raw $T_1$-weighted and dMRI data were processed accordingly. Namely, the dMRI data passed through an optimised pipeline[85]. The pipeline includes corrections for noise[91], Gibbs ringing[92], susceptibility-induced and motion distortions, and eddy current artifacts[93]. Isotropic 1 mm³ Gaussian smoothing was carried out using FSL's[94,95] *fslmaths*. Employing the multi-shell data, Diffusion Tensor Imaging (DTI)[96], Diffusion Kurtosis Imaging (DKI)[97] and White Matter Tract Integrity (WMTI)[98] metrics were estimated using Matlab 2017b code (https://github.com/NYU-DiffusionMRI/DESIGNER). Spherical mean technique (SMT)[99], and multi-compartment spherical mean technique (SMTmc)[100] metrics were estimated using original code (https://github.com/ekaden/smt)[99,100]. Estimates from the Bayesian Rotational Invariant Approach (BRIA) were evaluated by the original Matlab code (https://bitbucket.org/reisert/baydiff/src/master/)[101].

$T_1$-weighted images were processed using Freesurfer (version 5.3)[102] automatic *recon-all* pipeline for cortical reconstruction and subcortical segmentation of the $T_1$-weighted images (http://surfer.nmr.mgh.harvard.edu/fswiki)[103].

In total, we obtained 28 WM metrics from six diffusion approaches (DTI, DKI, WMTI, SMT, SMTmc, BRIA; see for overview in Supplement 9). In order to normalise all metrics, we used Tract-based Spatial Statistics (TBSS)[104], as part of FSL[94,95]. In brief, initially all brain-extracted[105] fractional anisotropy (FA) images were aligned to MNI space using non-linear transformation (FNIRT)[95]. Following, the mean FA image and related mean FA skeleton were derived. Each diffusion scalar map was projected onto the mean FA skeleton using the TBSS procedure. In order to provide a quantitative description of diffusion metrics we used the John Hopkins University (JHU) atlas[106], and obtained 30 hemisphere-specific WM regions of interest (ROIs) based on a probabilistic WM atlas (JHU)[107] for each of the 28 metrics. For $T_1$-weighted data, we applied the Desikan-Killiany Atlas[108]. Altogether, 840 dMRI features were derived per individual [28 metrics × (24 ROIs + 6 tracts)] for each hemisphere, and 117 $T_1$-weighted features (surface area, volume, thickness for each of the 34 regions; 3 whole-brain gray matter averages, and 2 averages of white matter surface area and volume) for each hemisphere.

### Brain age predictions
Brain age was predicted using the XGBoost algorithm[109] implemented in Python (v3.7.1). We used six data subsets to predict brain age split in the following manner: 1) right hemisphere $T_1$-weighted, 2) left hemisphere $T_1$-weighted, 3) left hemisphere diffusion, 4) right hemisphere diffusion, 5) left hemisphere multimodal, 6) right hemisphere multimodal. We applied nested $k$-fold cross-validation with 5 outer and 10 inner folds (see Supplementary Table 1 for tuned hyperparameters for

models trained on data from both sexes together and Supplementary Table 15 for models trained separately for males and females). We corrected for age-bias and mere age-effects[110,111] by including age in the regression equations (Eq. (5)) when assessing effects of modality, hemisphere, and handedness on brain age, as well as phenotype associations with brain ages (Eq. (9)).

**Statistical analyses**
All statistical analyses were carried out using Python (v3.7.1) and R (v4.2.0).

**Hemispheric differences and age sensitivity.** To give an overview of the extent of brain asymmetry, we assessed the significance of $T_1$-weighted and dMRI features' asymmetry using two-sided $t$-tests. The lateralisation or asymmetry of the brain features was estimated as the following: we applied the LI[32] to both regional features and features averaged over each hemisphere (see also ref. 33).

$$LI = \frac{L - R}{L + R}, \quad (1)$$

where $L$ and $R$ belongs to any left and right scalar metric, respectively. Furthermore, when associating LI with age, we used absolute LI values ($|LI|$) allowing to estimate age-effects on asymmetry irrespective of the direction of the asymmetry (leftwards or rightwards).

We then used linear regression models correcting for sex and scanning site to predict age from all regular and LI features:

$$\hat{Age} = F + Sex + Site, \quad (2)$$

where $F$ is a scalar metric such as, for example, hippocampus volume (derived from $T_1$-weighted image) or tapetum fractional anisotropy (derived from DTI). The same model setup was used applying generalised additive models (GAM) to model non-linear relationships between $F$ and $Age$ using a smooth $s$ of linked quadratic functions with $k = 4$ knots and restricted maximum likelihood (REML):

$$\hat{Age} = s(F) + Sex + Site. \quad (3)$$

Likelihood ratio tests (LRTs)[112] were used to assess the age sensitivity of all $T_1$-weighted and dMRI features and their asymmetry/LI features by comparing the above models with baseline models not including the respective feature:

$$\hat{Age} = Sex + Site. \quad (4)$$

We used the same procedure for region-averaged and hemispheric average metrics for regular and LI features. Hemispheric averages of regular features were then visualised by age, including surface area, volume, thickness for $T_1$-weighted data, and intra- and extra-axonal water diffusivities as well as for DTI and DKI metrics.

To compare the model fit of non-linear and linear models we used the Akaike information criterion (AIC)[113] and Bayesian information criterion (BIC)[114].

**Brain age assessment.** We estimated correlations across HBA and GBA to assess their similarities in addition to the model output provided from the prediction procedure. We also correlated age with the LI (see Eq. (1)) for the three modalities (dMRI, $T_1$-weighted, multimodal MRI), and estimated the age sensitivity of the LI as described in (Eqs. (2)–(4)).

As preregistered (https://aspredicted.org/if5yr.pdf), to test the relationships between hemisphere ($H$), modality ($M$), and HBA while controlling for age, sex, and scanner site, we employed linear mixed

effects regression (LMER) models of the following form:

$$\hat{HBA} = H + M + H \times M + Sex + Age + Sex \times Age + (1|Site) + (1|I), \quad (5)$$

where $I$ refers to the random intercept at the level of the individual. Post-hoc group differences were observed for hemisphere, modality and their interaction.

Next, handedness ($Ha$) was added to the model to observe whether there are model differences between the resulting LMER:

$$\hat{HBA} = Ha + H \times Ha + H + M + H \times M + Sex + Age + Sex \times Age + (1|Site) + (1|I), \quad (6)$$

and the previous model. Models were statistically compared using LRTs[112].

For sex-stratified analyses, we considered brain age estimates both from models using data from both sexes together, as well as models that were trained on females-only or males-only data. The modelling choice ($MC$) was included as a factor for the sex-stratified brain age analyses in the formula of Eq. (6).

Finally, the LIs (Eq. (1)) of left and right brain age predictions for $T_1$-weighted, diffusion and multimodal MRI ($LI_{HBA}$, i.e. the asymmetry in brain age predictions) were associated with age, controlling for sex and scanner site as random effect:

$$\hat{Age} = LI_{HBA} + Sex + (1|Site). \quad (7)$$

The $LI_{HBA}$s' age-sensitivity was then assessed (as for brain features, see Eqs. (2)–(4)), using LRTs comparing the above model with a baseline model excluding $LI_{HBA}$ (Eq. (4)):

$$\hat{Age} = Sex + (1|Site). \quad (8)$$

This procedure was also done for each sex individually, also separating between brain age models predictions which were obtained from the data from both sexes compared to a single sex.

**Phenotype associations of brain age.** In an exploratory analysis step, we assessed association patterns between brain ages and health and lifestyle factors which have previously demonstrated an association with brain age[20,26,38–41]. This analysis step served to compare phenotype associations across estimated brain ages. The health and lifestyle factors included alcohol drinking (binary), height and weight supplementing body mass index (BMI), diabetes diagnosis (binary), diastolic blood pressure, systolic blood pressure, pulse pressure, hypertension (binary), cholesterol level (binary), and smoking (binary describing current smokers). For this last analysis step, LMERs were used with the following structure:

$$\hat{P} = BA + Sex + Age + Sex \times Age + (1|Site), \quad (9)$$

where $BA$ refers brain age incorporating both GBA and HBA, $P$ is the phenotype.

Furthermore, where applicable, we corrected $p$-values for multiple testing using Bonferroni correction and an $\alpha$-level of $p < 0.05$. This involves multiplying the $p$-value by the number of tests used to test the same hypothesis. Adjusted $p$-values are marked as $p_{adj}$ and unadjusted $p$-values as $p$. We used a high-precision approach to calculate exact $p$-values utilising the Multiple Precision Floating-Point Reliable R package[115], and report standardised $\beta$-values. Sex and site were entered as independent factorial nominal variables in the applicable regression models, with sex being a binary (0 = female, 1 = male) and scanner site a multinominal (0 = Cheadle, 1 = Newcastle, 2 = Reading, 3 = Bristol). Finally, we repeated the presented statistical analyses stratifying for sex.

Article

**Effect size measures.** For the estimation of the effect size of group differences we used Cohen's *d*, estimated from the means of the two groups being compared ($\bar{X}_1, \bar{X}_2$) and the pooled standard deviation of the two groups (*s*):

$$d = \frac{\bar{X}_1 - \bar{X}_2}{s}.$$

For the estimation of the effect size of associations, we used the Pearson correlation coefficient, which is estimated from two variables' individual data points ($x_i, y_i$), and their averages ($\bar{x}, \bar{y}$):

$$r = \frac{\sum(x_i - \bar{x})(y_i - \bar{y})}{\sqrt{\sum(x_i - \bar{x})^2 \cdot \sum(y_i - \bar{y})^2}}.$$

For multivariate associations, we used regression coefficients / *β*-weights from the respective regression equation (see Eq. (2)–(9)).

### Reporting summary
Further information on research design is available in the Nature Portfolio Reporting Summary linked to this article.

## Data availability
This study has been conducted using UKB data under Application 27412. All raw data are available from the UKB (www.ukbiobank.ac.uk). UK Biobank has approval from the North West Multi-centre Research Ethics Committee (MREC) as a Research Tissue Bank (RTB) approval.

The raw and processed UK Biobank MRI data are protected and are not openly available due to data privacy laws. However, access can be obtained by applying for access and paying an access fee (see https://www.ukbiobank.ac.uk/enable-your-research/apply-for-access). Source data are provided with this paper.

## Code availability
Analysis code[116] is available at https://doi.org/10.5281/zenodo.10423745.

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

## Acknowledgements

This study has been conducted using UKB data under Application 27412. UKB has received ethics approval from the National Health Service

National Research Ethics Service (ref 11/NW/0382). The work was performed on the Service for Sensitive Data (TSD) platform, owned by the University of Oslo, operated and developed by the TSD service group at the University of Oslo IT-Department (USIT). Computations were performed using resources provided by UNINETT Sigma2 - the National Infrastructure for High Performance Computing and Data Storage in Norway.

We want to thank Tobias Kaufmann and Torgeir Moberget who processed the $T_1$-weighted MRI data, and all UKB participants and facilitators who made this research possible.

## Author contributions

M.K.: Study design, Software, Formal analysis, Visualizations, Project administration, Writing—original draft, Writing—review & editing. D.v.d.M.: Software, Writing—review & editing. D.B.: Writing—review & editing. A.M.G.d.L.: Software, Writing—review & editing. A.L.: Funding acquisition. E.E.: Funding acquisition. O.A.A.: Writing—review & editing, Funding acquisition. L.T.W.: Writing—review & editing, Funding acquisition. I.I.M.: Supervision, Study design, Data preprocessing, and quality control, Writing—review & editing, Funding acquisition.

## Funding

This research was funded by the Research Council of Norway (#223273, L.T.W.; #324252, O.A.A.); the South-Eastern Norway Regional Health Authority (#2022080, O.A.A.); the European Union's Horizon2020 Research and Innovation Programme (#847776, O.A.A.; #802998 L.T.W.); the Swiss National Science Foundation (#PZ00P3_193658, A.M.G.d.L); and the Trond Mohn Foundation (Grant BFS2018TMT07, A.L.). Open access funding provided by Western Norway University Of Applied Sciences.

## Competing interests

The authors declare the following competing interests: O.A.A. has received a speaker's honorarium from Lundbeck and is a consultant to Coretechs.ai. The remaining authors declare no other competing interests.
