## [Peer Review File · Nature Communications]

Brain asymmetries from mid- to late life and hemispheric brain ageREVIEWER COMMENTS

Reviewer #1 (Remarks to the Author):

The article "Brain asymmetries from midlife to old adulthood and hemispheric brain age" is one of the most comprehensive articles on this topic due to large sample data, multimodal datasets, multiple metrics and different brain age prediction models used. The study is an observational one, looking at brain morphology in relation with age, considering the brain asymmetry. The analyses reported were global, at the level of hemispheres and also regional. In general, the asymmetries appear lower at higher age for grey and white matter, which means that the difference in left/right hemispheric brain age is smaller late in life. The findings are supported by literature, but the novelty comes from pooling together many modalities. The regional WM asymmetry for example outlines fornix-striaterterminalis asymmetry to decrease and cingulum asymmetry to increase with age. The pool of participants in general is supposed to be healthy subjects, however subtle tendencies/predispositions to different illnesses might not be very well documented. The authors considered this in their discussion and interpret the asymmetry in relation with potential vulnerabilities. The limitations should be clearly stated in the study.

The article is easy to read, the messages easy to get and it can be a big contribution in the field. A very minor thing I would avoid using the word "old" for "late-life" or "elderly".

Reviewer #2 (Remarks to the Author):

Recently, technological advancements have made it possible to estimate brain age using MRI scans. Brain age is computed based on the appearance of the brain MRI, but it does not always match the actual age due to various reasons that can cause the brain to age faster or slower. This new technology could be valuable for exploring brain health through data analysis, but it has not yet been studied in the context of asymmetries. In their insightful research paper, titled "Brain asymmetries from midlife to old adulthood and hemispheric brain age," Korbmacher et al. investigate anatomical brain asymmetries and brain age asymmetry changes cross-sectionally with age.

The report is well-written, but I have some major recommendations.

1. The study is cross-sectional, meaning the participants were born in different decades, which may impact their brain health due to various environmental, educational, or cultural reasons. While I understand that there is no other way to explore this question, the report should acknowledge the potential contamination of the results by these other factors as a limitation.

2. When correcting for sex and scanning site, the report should specify how these variables were entered in the linear regression. Were there two variables (site and sex) or one variable for males (binary 0 or 1), one variable for females (binary 0 or 1), and one variable per site?

3. While it is important to remove the variance related to sex by means of linear regression to provide general results independent of the impact of sex, the effect of sex in the analysis is fundamental to many modern debates and should be reported in the manuscript.

4. Many manuscripts indicate that the effect of aging on neuroimaging variables is non-linear, which is also evident from Figure 1 in your manuscript. The report used linear mixed-effects regression to assess statistical significance, but the non-linearity of the effects hinders this method. Adjusting the model or transforming the data might provide more accurate results.

Reviewer #3 (Remarks to the Author):

This study proposed a metric of hemispheric brain age (HBA) to investigate brain asymmetry using T1- and diffusion-MRI data from the UK Biobank. Although the GM and WM features used to compute brain age showed some extent of asymmetry, the estimated brain age did not vary significantly across hemispheres, nor vary across modalities or handedness. They also investigated the relationship between brain age and several health-and-lifestyle factors but found no significant difference between HBA and global brain age (GBA) estimates. However, they found a negative correlation between brain age asymmetry and chronological age, which is consistent with brain asymmetry estimated using raw brain MRI features.

This is a descent study, and the manuscript is written well.

However, this study overall offers limited new insight into i) brain asymmetry, compared to known knowledge reflected by raw MRI GM and WM features; and ii) the relationship between brain aging and health and disease, compared to established global brain age estimates.

Comments relating to Methods/Results:

The model fitting shown in Figure 1 (and several related SI figures) is confusing and appears overfitted. It is unclear how each model was fitted. The larger variation shown at the two ends of the age range may be due to reduced sample size. Although it is claimed not to be case in the Discussion, no proper test was performed to support this claim. Also, the abbreviation of the name of the phenotype shown in each panel needs to be defined in the caption to improve the readability.

Table 1 presented a metric called corrected correlation between predicted and age. However, it is unclear why this so-called corrected correlation is needed and how was this metric computed. It is unclear whether the predicted accuracy described in the manuscript refers to the raw correlation or the corrected correlation

Responses to the Reviewers' Comments

General response: We would like to express our sincere gratitude to the three Reviewers for their insightful and encouraging comments. Our point-by-point response is detailed below in blue. Changes to the manuscript are also marked in blue.

Reviewer #1:

The article “Brain asymmetries from midlife to old adulthood and hemispheric brain age” is one of the most comprehensive articles on this topic due to large sample data, multimodal datasets, multiple metrics and different brain age prediction models used. The study is an observational one, looking at brain morphology in relation with age, considering the brain asymmetry. The analyses reported were global, at the level of hemispheres and also regional. In general, the asymmetries appear lower at higher age for grey and white matter, which means that the difference in left/right hemispheric brain age is smaller late in life. The findings are supported by literature, but the novelty comes from pooling together many modalities. The regional WM asymmetry for example outlines fornix-striaterminalis asymmetry to decrease and cingulum asymmetry to increase with age. The pool of participants in general is supposed to be healthy subjects, however subtle tendencies/predispositions to different illnesses might not be very well documented. The authors considered this in their discussion and interpret the asymmetry in relation with potential vulnerabilities. The limitations should be clearly stated in the study. The article is easy to read, the messages easy to get and it can be a big contribution in the field. A very minor thing I would avoid using the word “old” for “late-life” or “elderly”.

Author Response: We want to thank the Reviewer for this positive feedback. We have now altered the text at respective places in order to avoid the usage of the word “old” to describe the late adulthood. This includes a change to the title, which now reads “Brain asymmetries from mid- to late life and hemispheric brain age”. We have also added a dedicated paragraph discussing several limitations of our study and the UK Biobank (UKB) imaging sample:

“While this study provides initial explorations of asymmetries and HBA, our findings remain limited to the examined sample (imaging subset of the UKB). The UKB contains individuals born in different decades, which influences individual predispositions for brain health through various factors such as the living environment or education, representing various potential confounding effects. Additional bias might have been introduced by the sample characteristics and sampling procedure. The UKB consists of nearly exclusively white UK citizens, limiting the generalisability beyond white Northern Europeans and US Americans in their midlife to older age. The volunteer-based sampling procedure might additionally have introduced bias, reducing generalisability to the UK population, with the imaging sample of the UKB showing an additional

positive health bias (better physical and mental health) over the rest of the UKB sample, rendering this sub-sample as even less representative of the total UK population.” (pp.12-13, ll.450-462).

Reviewer #2:

Recently, technological advancements have made it possible to estimate brain age using MRI scans. Brain age is computed based on the appearance of the brain MRI, but it does not always match the actual age due to various reasons that can cause the brain to age faster or slower. This new technology could be valuable for exploring brain health through data analysis, but it has not yet been studied in the context of asymmetries. In their insightful research paper, titled “Brain asymmetries from midlife to old adulthood and hemispheric brain age,” Korbmacher et al. investigate anatomical brain asymmetries and brain age asymmetry changes cross-sectionally with age.

The report is well-written, but I have some major recommendations.

1. The study is cross-sectional, meaning the participants were born in different decades, which may impact their brain health due to various environmental, educational, or cultural reasons. While I understand that there is no other way to explore this question, the report should acknowledge the potential contamination of the results by these other factors as a limitation.

Author Response: Thank you for pointing out this limitation to the presented findings. We have now included a paragraph in the end of the Discussion section addressing these issues. The paragraph reads:

”While this study provides initial explorations of asymmetries and HBA, our findings remain limited to the examined sample (UKB), and limited by generational effects within the sample. The UKB contains individuals born in different decades, which influences individual predispositions for brain health through various factors such as the living environment or education, representing various potential confounding effects. Additional bias might have been introduced by the sample characteristics and sampling procedure. The UKB consists of nearly exclusively white UK citizens, limiting the generalisability beyond white Northern Europeans and US Americans in their midlife to late life. The volunteer-based sampling procedure might additionally have introduced bias, reducing generalisability to the UK population, with the imaging sample of the UKB showing an additional positive health bias (better physical and mental health) over the rest of the UKB sample, rendering this sub-sample as even less representative of the total UK population.” (pp.12-13, ll.450-462).

2. When correcting for sex and scanning site, the report should specify how these variables were entered in the linear regression. Were there two variables (site and sex) or one variable for males (binary 0 or 1), one variable for females (binary 0 or 1), and one variable per site?

Author Response: We have now clarified the explanation of how sex and scanner site were entered in the regression models at the end of the Methods section:

”Sex and site were entered as independent nominal, factorial variables in the applicable regression models, with sex being a binary (0 = female, 1 = male) and scanner site a multinomial (0 = Cheadle, 1 = Newcastle, 2 = Reading, 3 = Bristol).” (p.17, ll.574-578)

3. While it is important to remove the variance related to sex by means of linear regression to provide general results independent of the impact of sex, the effect of sex in the analysis is fundamental to many modern debates and should be reported in the manuscript.

Author Response: We agree with the reviewer that characterising the effects of sex is important and should be reported. Additionally, the Nature Communications reporting guidelines require sex-stratifications and we have hence included additional analyses stratifying by sex for all previously reported analyses, as well as sex-interaction terms with hemisphere and modality when examining hemispheric brain age estimates. The new analyses are now specified in the Methods section, throughout the Results sections, and in two new sex-specific Results sections, and briefly discussed throughout the Discussion section. The two new sections in the Results section read ”Sex stratified hemispheric differences and age sensitivity for GM and WM features” (p.7-8) and ”Sex differences in GM and WM feature asymmetry” (p.8-9).

Importantly, these analyses presented crucial differences for brain age estimates between males and females confirming our initial hypotheses on the influence of hemisphere, modality and handedness on brain age:

”These results were robust to stratifying by sex, estimates from a brain age model considering both sexes for unadjusted ($r_{dMRI\ males} = -0.134$, $r_{dMRI\ females} = -0.104$, $r_{T1\ males} = -0.134$, $r_{T1\ females} = -0.048$, $r_{multimodal\ males} = -0.134$, $r_{multimodal\ females} = -0.111$), and adjusted associations ($\beta_{dMRI\ males} = -0.134$, $\beta_{dMRI\ females} = -0.099$, $\beta_{T1\ males} = -0.134$, $\beta_{T1\ females} = -0.045$, $\beta_{multimodal\ males} = -0.134$, $\beta_{multimodal\ females} = -0.106$), with χ^2 tests suggesting age sensitivity (all $p < .001$). Using brain age predictions from models which were independently estimated for males and females showed similar results for unadjusted ($r_{dMRI\ males} = -0.141$, $r_{dMRI\ females} = -0.094$, $r_{T1\ males} = -0.120$, $r_{T1\ females} = -0.031$, $r_{multimodal\ males} = -0.165$, $r_{multimodal\ females} = -0.089$), and adjusted associations ($\beta_{dMRI\ males} = -0.137$, $\beta_{dMRI\ females} = -0.088$, $\beta_{T1\ males} = -0.117$, $\beta_{T1\ females} = -0.029$, $\beta_{multimodal\ males} = -0.162$, $\beta_{multimodal\ females} = -0.084$), with χ^2 tests suggesting age sensitivity (all $p < .001$). Finally, also when analysing brain ages for males and females from sex-specific models together shows similar trends for uncorrected $|LI_{HBA}|$ -age associations ($r_{multimodal} = -0.123$, $p < .001$; $r_{T1} = -0.074$, $p < .001$, $r_{dMRI} = -0.114$, $p < .001$), as well as corrected association

($\beta_{multimodal} = -0.125$, $p < .001$; $\beta_{T_1} = -0.071$, $p < .001$, $\beta_{dMRI} = -0.113$, $p < .001$; Eq. 7-8)." (pp.6-7, ll.178-205).

Moreover, we added information on the interaction between sex and hemisphere and between sex and modality by estimating marginal effects when closer examining the hemispheric brain age predictions in the Section "Sex-specific differences in the influence of hemisphere, modality, and handedness on brain age estimates", p. 5-6, ll.142-176.

The implications of these findings are further specified in the Discussion section:

"Brain age predictions exhibited concordant accuracy within modalities for left, right, and both hemispheres, and concordant associations with health-and-lifestyle factors also when analysing data for males and females separately when training the model on data from both sexes. The predictions did not differ statistically between hemispheres, modalities, or handedness groups when considering both sexes together. However, sex-stratified analyses revealed opposing effects between sexes for hemisphere and modality but not handedness. There are multiple reasons for the observed higher brain age in females' right hemisphere compared to males' higher brain age of the left hemisphere, in addition to modality-specific differences. First, male and female brain structure differs, resulting in sex-specific regional variations in brain age estimates. Second, body and brain ageing trajectories differ between sexes, for example, outlined by sex-dependent importance of cardiometabolic risk factors. Hence, the tendency of males' predicted brain age being lower using T₁-weighted and multimodal in contrast to diffusion-derived brain ages, with these trends reversed in females, might also reflect stronger brain age associations with cardiometabolic risk factors in males (Supplementary Figure 7), which have been demonstrated earlier for WM features and WM brain age. HBA allows one to assess the structural integrity of each hemisphere individually, and to set brain ages from the two hemispheres in relationship to each other providing a general marker of asymmetry. Despite brain asymmetries overall increasing (Supplementary Figures 10-11), the asymmetry between left/right HBA were smaller at a higher age. At higher ages, both hemispheres might, hence, become overall more comparable, despite ageing-related changes".(p.9, ll.298-321).

And: "[...] yet different between hemispheres and modalities within males and females, with these differences contrasting each other. This renders HBA sensitive to potential underlying biological processes which only become apparent when assessing males and females separately. Additionally, different modalities might be sensitive to a range of biological phenomena in terms of brain age, such as dMRI brain age which is correlated with diabetes only in males. In that sense, a further route of investigation could be to establish sex-specific uni- and multimodal brain age models (which account for sex differences in brain morphology and its developmental trajectories). The influence of hemisphere and sex on how these models relate to biological phenomena can then be assessed." (p.12, ll.429-438).

Finally, sex differences were however small for brain feature asymmetries, the relationship of brain feature asymmetries and age, as well as the relationship between brain age asymmetry and age (see pp.5-8).

4. Many manuscripts indicate that the effect of aging on neuroimaging variables is non-linear, which is also evident from Figure 1 in your manuscript. The report used linear mixed-effects regression to assess statistical significance, but the non-linearity of the effects hinders this method. Adjusting the model or transforming the data might provide more accurate results.

Author Response: This is an excellent comment, we agree with the Reviewer's comment. We have included non-linear models in the form of generalized additive models (GAM) using a smooth of linked quadratic functions with $k = 4$ knots and restricted maximum likelihood for all relevant analyses, particularly in the sections "Hemispheric differences and age sensitivity for GM and WM features", and the new section "Sex stratified hemispheric differences and age sensitivity for GM and WM features" (pp.7-9), in addition to smaller changes in the other sections.

Furthermore, we compared linear and non-linear models by comparing the models on the Akaike and Bayesian information criterion to gain a better understanding of the suitability of linear compared to non-linear models when examining the relationship of age and brain asymmetries in the present sample (p.5, ll.113-120 for analyses including both sexes; p.8, ll.247-257 for sex-stratified analyses). The comparisons suggests no significant difference in model fit. As we presented the slopes of hundreds of region-specific brain features' asymmetries with age, we deemed the slopes of such relationships as a useful criterion of comparison. As non-linear models do not possess constant slopes, we have kept the presentation of slopes (as a measure of effect size between age and features' LI), in addition to the aggregate statistics of the slopes, as previously presented.

Reviewer #3:

This study proposed a metric of hemispheric brain age (HBA) to investigate brain asymmetry using T1- and diffusion-MRI data from the UK Biobank. Although the GM and WM features used to compute brain age showed some extent of asymmetry, the estimated brain age did not vary significantly across hemispheres, nor vary across modalities or handedness. They also investigated the relationship between brain age and several health-and-lifestyle factors but found no significant difference between HBA and global brain age (GBA) estimates. However, they found a negative correlation between brain age asymmetry and chronological age, which is consistent with brain asymmetry estimated using raw brain MRI features. This is a descent study, and the manuscript is written well. However, this study overall offers limited new insight into i) brain asymmetry, compared to known knowledge reflected by raw MRI GM and WM features; and ii) the relationship between brain aging and health and disease, compared to established global brain age estimates.

Comments relating to Methods/Results:

The model fitting shown in Figure 1 (and several related SI figures) is confusing and appears overfitted. It is unclear how each model was fitted.

Author Response: Thank you for making us aware of this lack of information. Model characteristics are now clearly stated for each presented figure. Previously, we had used $k = 80$ knots only restricting the spline functions (standard setting in R ggstats and ggplot related packages). To avoid overfitting, we have now restricted all (cubic) spline functions to $k = 4$ knots (see Figure 1: p.28, and Supplementary Figures: p.44, pp.51-62).

Instead of using the linear models to correct the different values and afterwards fitting smooth curves in Figure 1, we have now applied generalised additive models based on the suggestions of Reviewer 2 in the case of this figure and all other age associations throughout the manuscript, presenting the average marginal effect/slope of age on the different metrics. In order to offer simple insights based on a linear fit, we kept them in the text as well.

The larger variation shown at the two ends of the age range may be due to reduced sample size. Although it is claimed not to be case in the Discussion, no proper test was performed to support this claim.

Author Response: We agree with the Reviewer and have now removed this claim from the Discussion section.

Also, the abbreviation of the name of the phenotype shown in each panel needs to be defined in the caption to improve the readability.

Author Response: Thank you for pointing this out. The figure caption is now updated to aid readability both in caption and by adding black frames for significant associations. The caption now reads "Linear association between general health-and-lifestyle phenotypes and brain age estimated from different modalities, left, right and both hemispheres. Eq. 9 was used and standardized slopes are presented. For simplicity, standardized slopes with $|\beta| < 0.005$ were rounded down to $|\beta| = 0$. L: left hemisphere, R: right hemisphere, LR: both hemispheres, BMI: body mass index, WHR: waist-to-hip ratio. Bonferroni-adjusted $p < .05$ is marked by a black frame." (pp.31, 57, 61-62)

Table 1 presented a metric called corrected correlation between predicted and age. However, it is unclear why this so-called corrected correlation is needed and how was this metric computed. It is unclear whether the predicted accuracy described in the manuscript refers to the raw correlation or the corrected correlation.

Author Response: We agree that a corrected brain age metric (the linear age-residualised brain age) does not contribute in the case of this manuscript as these estimates inflate the age-relationship and at the same time do not

contribute any additional information on the modelling. We have now removed the corrected metric from the manuscript.

Additional Changes:

After careful consideration, we have now included *absolute* laterality indexed features' age associations, indicating the age association of general asymmetry per feature. These associations are more comparable across features than previous non-absolute LI-based associations, as absolute LIs allow for all intercepts being positive compared to previous intercepts being both positive and negative (indicating left-ward and rightward asymmetry). Hence, absolute LI estimates allow for a clear interpretation of meta-statistics indicating on average higher white and grey matter asymmetry at a higher age. Resulting changes across the manuscript are detailed below.

Methods section:

We include a short description of the absolute LI: "Furthermore, when associating LI with age, we used absolute LI values ($|LI|$) allowing to estimate age-effects on asymmetry irrespective of the direction of the asymmetry (leftwards or rightwards)" (p.15, ll.544-547).

Results section:

We now report absolute LI for analyses concerning features and brain age. These changes to the analysis strategy apply also to the new sex-stratified analyses (pp.5-6, pp.8-9).

Discussion section:

We adapted the Discussion section based on the new observation of higher absolute brain feature asymmetry (but lower brain age asymmetry) at higher ages.

"Brain asymmetry was age-sensitive, and overall higher at higher ages. In contrast, asymmetry in hemispheric brain age was lower at higher ages. The strongest relationship of age and absolute brain asymmetry was identified in larger GM and WM regions, as well subcortical structures, including the limbic system, the ventricles, cingulate and cerebral as well as cerebellar peduncle WM." (p.8, ll.293-297)

"We extended previous findings by providing a comprehensive overview of brain asymmetry associations throughout mid- to late life including both GM and WM. Our findings indicate that when considering various metrics, older brains generally appear less symmetric than younger brains in the current sample mid- to late life sample, whereas brain age appears more symmetric in older brains." (p.10, ll.362-366).

"Notably, we identified strong associations between specific brain regions' asymmetry and age. The strongest age-associations of asymmetries were observed for subcortical, ventricle-near structures. The general age-sensitivity of

such structures might be a reason for the observed age-associations in asymmetries, and hence pointing towards one hemisphere being stronger affected by degradation effects, or even the involvement of such regions in psychiatric and neurodegenerative disorders. For example, the hippocampus, a prominent limbic structure, presents relatively high levels of adult neurogenesis, which might potentially explain repeated findings of the region's associations with psychiatric disorders and disorder states such as depression, anxiety, schizophrenia, addiction, and psychosis, and neurodegenerative disorders, especially Alzheimer's Disease, but also ageing in general. Some of the strongest age-relationship for T₁-derived asymmetries were observed in accumbens, ventricles and pallidum. In turn, a series of dMRI approaches was sensitive to asymmetry in the cingulum tract, which is higher in late-life and peduncle asymmetry which appears lower in late-life. In particular radial diffusivity metrics, such SMT-trans, SMTmc-extratrans, and BRIA-microRd, and fractional anisotropy indicated by BRIA-microFA were sensitive to age-dependencies of these asymmetries. Although speculative, this observation could indicate a relationship between asymmetry and axonal properties during ageing, such as myelination, density, or diameter, in the cingulum, with yet a more general marker (BRIA-microFA) of anisotropy asymmetry increasing at advanced age. However, limitations of the different diffusion metrics, such as the inability to account for axonal swelling, infection, or crossing fibres, aggravates the interpretation of such asymmetry changes. Overall, asymmetries' age-dependencies in subcortical, limbic and ventricle-near areas are not surprising, considering that the cingulum and cerebral peduncle WM, and middle temporal GM area also presented some of the strongest asymmetries across the sample (Supplementary Table 10). Both GM volume, surface, and thickness show asymmetries across studies. We identified lower asymmetry linked to higher ages in the ventricular and pallidum volumes, appearing alongside the known effect of larger ventricle volumes at higher ages. The strongest positive age-relationships for T₁-weighted features' asymmetry were observed for accumbens and WM surface area, as well as limbic structures such as amygdala, hippocampus, and cingulate. Limbic structures have previously been outlined as highly age-sensitive. Higher asymmetry-levels might speak to asymmetric atrophy in these limbic regions, potentially explaining several ageing-related effects. However, lifespan changes in ventricular volume asymmetry in relation to symptom and disorder expression requires additional investigations." (p.11, ll.394-402).

REVIEWERS' COMMENTS

Reviewer #1 (Remarks to the Author):

The authors have adequately addressed my comments.

Reviewer #2 (Remarks to the Author):

This is an excellent revision. I congratulate the authors for a job well done. I have no further comments.

Reviewer #4 (Remarks to the Author):

Korbmacher and co-authors have conducted a comprehensive study on brain asymmetries during mid- to late-life, focusing on the UK Biobank cohort. Their research explores inter-hemispheric differences in regional metrics across various modalities, and their correlations with age. By utilizing a brain age and hemispheric brain age approach, the study reveals smaller disparities in hemispheric brain age among older individuals. Furthermore, the authors report intriguing sex-specific differences in both regional association and brain age estimates. Overall, this work provides valuable data for understanding brain asymmetry and its role in the aging process. I have a few minor comments.

The manuscript encompasses a multitude of tests examining hemispheric differences in various regions and metrics, along with their associations with age, across different subgroups. Although the study presents extensive findings, it would be beneficial for the authors to include information on multiple testing correction to ensure the robustness of their results. Providing further clarification on this issue would enhance the overall comprehensibility of the research.

In reference to the observed hemispheric differences derived from cortical parcellations and white tracts, it is essential to consider the possibility that these differences may be influenced by potential parcellation bias rather than solely reflecting inherent brain organization.

Please double check reference 8 for its support for the association between functional network difference and handedness in the first Introduction paragraph.

Please double check reference 4 for its link with 'obsessive-compulsive disorder' studies.

Responses to the Reviewers' Comments

General response: We want to thank the four Reviewers for reviewing this manuscript, which ultimately lead to substantial improvements. A final point-by-point response is detailed below in blue. Changes to the manuscript are also marked in blue in the manuscript file.

Reviewer #1:

The authors have adequately addressed my comments.

Reviewer #2:

This is an excellent revision. I congratulate the authors for a job well done. I have no further comments.

Reviewer #3:

No comments provided.

Reviewer #4:

Korbmacher and co-authors have conducted a comprehensive study on brain asymmetries during mid- to late-life, focusing on the UK Biobank cohort. Their research explores inter-hemispheric differences in regional metrics across various modalities, and their correlations with age. By utilizing a brain age and hemispheric brain age approach, the study reveals smaller disparities in hemispheric brain age among older individuals. Furthermore, the authors report intriguing sex-specific differences in both regional association and brain age estimates. Overall, this work provides valuable data for understanding brain asymmetry and its role in the aging process. I have a few minor comments.

The manuscript encompasses a multitude of tests examining hemispheric differences in various regions and metrics, along with their associations with age, across different subgroups. Although the study presents extensive findings, it would be beneficial for the authors to include information on multiple testing correction to ensure the robustness of their results. Providing further clarification on this issue would enhance the overall comprehensibility of the research.

Author Response: We want to thank the Reviewer for the positive feedback and the useful comments. The implementation of the multiple comparison correction procedure (Bonferroni) is now further specified in the Methods section to separate unadjusted from adjusted p-values and the reasoning behind the adjustment throughout the manuscript:

"Furthermore, where applicable, we corrected p -values for multiple testing using Bonferroni correction and an α -level of $p < .05$. This involves multiplying

the p -value by the number of tests used to test the same hypothesis. Adjusted p -values are marked as p_{adj} and unadjusted p -values as p ." (p. 17, ll. 578-580).

In reference to the observed hemispheric differences derived from cortical parcellations and white tracts, it is essential to consider the possibility that these differences may be influenced by potential parcellation bias rather than solely reflecting inherent brain organization.

Author Response: We agree with the Reviewer's comment and have now added another sentence to the Discussion section mentioning this issue: "Finally, the selection of the parcellation scheme for both grey and white matter, and the inherent parcellation bias, might have influenced the results. As a consequence, the reported findings might be more reflective of the parcellation bias than the inherent brain organisation." (p. 13, ll. 464-467)

Please double check reference 8 for its support for the association between functional network difference and handedness in the first Introduction paragraph. Please double check reference 4 for its link with 'obsessive-compulsive disorder' studies.

Author Response: We have removed reference 4 at the respective place, and changed the sentence mentioning reference 8: "Furthermore, handedness has been repeatedly assessed together with asymmetry in humans and animals, and relates to brain asymmetry" (p. 2, ll. 5-7)